# Advanced MRI Techniques: Diagnosis and Follow-Up of Multiple Sclerosis

**DOI:** 10.3390/diagnostics14111120

**Published:** 2024-05-28

**Authors:** Riccardo Nistri, Antonio Ianniello, Valeria Pozzilli, Costanza Giannì, Carlo Pozzilli

**Affiliations:** 1Department of Human Neuroscience, Sapienza University, 00185 Rome, Italy; antonio.ianniello@uniroma1.it (A.I.); costanza.gianni@uniroma1.it (C.G.); carlo.pozzilli@uniroma1.it (C.P.); 2Fondazione Policlinico Universitario Campus Bio-Medico, 00128 Rome, Italy; 3Unit of Neurology, Neurophysiology, Neurobiology and Psychiatry, Department of Medicine and Surgery, Università Campus Bio-Medico di Roma, 00128 Rome, Italy; 4IRCCS Neuromed, 86077 Pozzilli, Italy; 5MS Center Sant’Andrea Hospital, 00189 Rome, Italy

**Keywords:** multiple sclerosis, MRI, advance imaging, chronic inflammation

## Abstract

Brain and spinal cord imaging plays a pivotal role in aiding clinicians with the diagnosis and monitoring of multiple sclerosis. Nevertheless, the significance of magnetic resonance imaging in MS extends beyond its clinical utility. Advanced imaging modalities have facilitated the in vivo detection of various components of MS pathogenesis, and, in recent years, MRI biomarkers have been utilized to assess the response of patients with relapsing–remitting MS to the available treatments. Similarly, MRI indicators of neurodegeneration demonstrate potential as primary and secondary endpoints in clinical trials targeting progressive phenotypes. This review aims to provide an overview of the latest advancements in brain and spinal cord neuroimaging in MS.

## 1. Introduction

Multiple Sclerosis (MS) is an autoimmune demyelinating disease involving the central nervous system (CNS) [1]. An estimated 2 million people are currently affected by the disease worldwide [2], with MS standing as the predominant incapacitating neurological disorder among young adults, typically manifesting its symptoms between the ages of 20 and 40 years [3]. It is a chronic condition that can present in different ways among individuals. Whilst some experience a benign trajectory with minimal disability, others face a progressively deteriorating course leading to escalating impairment over time [4,5]. Yet for most, the disease unfolds in episodic bouts of symptoms followed by prolonged periods of quiescence, often with partial or complete recovery. Fatal outcomes are rare, and individuals with MS generally maintain a normal life expectancy [5,6].

MS diagnosis and its distinctions from other demyelinating diseases or MS mimics can be challenging [7]. The current diagnostic criteria, 2017 McDonald criteria, use a combination of clinical, radiological, and laboratory features to guide clinicians [8]. In particular, pivotal elements in the MS diagnosis are represented by dissemination in space (DIS), defined as at least two different areas of CNS involved during clinical attacks or detected at the magnetic resonance imaging (MRI), and dissemination in time (DIT), represented by the appearance of new clinical attacks, the detection of new lesions during an MRI, or by the presence of oligoclonal bands in the cerebrospinal fluid (CSF) [9].

Different phenotypes of MS have been distinguished according to the course of the disease, whether inflammatory or progressive: relapsing–remitting MS (RRMS), secondary progressive MS (SPMS), and primary progressive MS (PPMS) [10]. Clinically isolated syndrome (CIS) and radiologically isolated syndrome (RIS), based on their features, can also be considered to be part of a wider MS phenotype classification [11].

In this context, MRI is central to the detection of new inflammatory activity, characterized by the appearance of new fluid-attenuated inversion recovery (FLAIR) (Figure 1) or new T2 lesions, the observation of T2 enlarging lesions, or the detection of new contrast-enhancing lesions in T1w sequences, a sign of acute inflammation [12].

In clinical investigations, conventional MRI has served to confirm diagnoses and evaluate the efficacy of disease-modifying treatments (DMTs) by quantifying new lesion formation and volumes [13,14]. Robust correlations have been established between the impact of new DMTs on new lesion formation and relapse rates, aiding in the exploration of the efficacy of novel therapies in phase II trials [15,16]. The term “no evidence of disease activity”, comprising the absence of clinical relapses, progression on the expanded disability status scale (EDSS), and MRI activity (e.g., new or enlarging T2 lesions, gadolinium-enhancing lesions, and potential brain atrophy), has been pivotal in assessing treatment outcomes, extending from clinical trials to practical clinical settings [17,18].

The presence of T1 hypointensities, known as “black holes”, signifies substantial axonal loss and correlates with heightened disability [19,20]. Advanced imaging techniques, such as whole-brain or gray matter volumetry, have emerged as vital secondary endpoints in clinical trials [15]. Despite ongoing improvements in imaging methodologies, conventional imaging modalities harbor limitations, perpetuating the clinical/radiological paradox wherein “normal-appearing” white and gray matter still exhibit detectable abnormalities via advanced techniques [21,22].

Innovative imaging techniques aim to unveil the neuroinflammatory and neurodegenerative aspects of MS not discernible through conventional means. These methodologies hold promise for enhanced sensitivity as biomarkers of disease activity and progression, with potential applications as surrogate endpoints in therapeutic trials [21,23].

This review aims to give clinicians an overview of the different methods used today, mainly in research settings, and the contribution they can potentially make in the future to MS diagnosis and follow-up.

## 2. Cortical Lesions

MS manifests in both white and grey matter and it is characterized by sharply delimited lesional areas featuring primary demyelination, axonal loss, and reactive gliosis [24]. This pathological process is generalized throughout the entire central nervous system and detectable in both white and grey matter lesions [24,25]. Though research and clinical practice have traditionally focused on white matter lesions (WMLs), it is now widely recognized that cortical damage is a significant aspect of the disease [24,26], as different studies indicate that cortical lesions (CLs) exhibit a stronger correlation with the severity of cognitive and physical disabilities compared to WMLs, and CLs have also been incorporated in the 2017 McDonald criteria [27,28]. Four types of cortical lesions have been described: leukocortical, intracortical, subpial, and subpial extending the entire width of the cortex (types 1–4, respectively) [29].

While common MRI protocols typically encompass T1-weighted (T1w), T2-weighted (T2w), and FLAIR [30], sequences with enhanced sensitivity for CLs detection, such as magnetization-prepared 2 rapid acquisition with gradient echo (MP2RAGE) [31,32] and double inversion recovery (DIR) [33,34], are increasingly employed in clinical practice [35]. Accurate lesion identification is crucial for monitoring disease activity and assessing the effectiveness of DMTs [36]. The number of CLs changes according to the phase of the disease, as they are more prevalent in SPMS compared to CIS or RRMS [37], with an impact on disability and cognitive impairment, thereby being vital indicators of disease progression in MS patients [27,28]. However, due to their small size and low contrast relative to normal grey matter, cortical lesions are inconspicuous on images acquired using conventional clinical MRI sequences (e.g., T1w, T2w, or FLAIR) [38]. Advanced MRI techniques, including DIR, MP2RAGE, and phase-sensitive inversion–recovery (PSIR) sequences, have significantly improved the visualization of cortical lesions, the latter appearing as a hyperintense signal (in DIR) or as a hypointense lesions (MP2RAGE and PSIR represented in Figure 2); however, such techniques find limited use in routine diagnostic and clinical trial evaluation protocols due to their prolonged acquisition times (i.e., 10–15 min) [31,34,38]. Moreover, CLs are primarily visible on advanced MRI sequences at high (3T) and ultra-high (7T) magnetic fields [27,28], which are not routinely available in clinics due to their high cost and their use is often limited to the research practice.

Pilot studies leveraging artificial intelligence (AI) have recently facilitated the generation of synthetic DIR images from standard clinical MRI protocols (i.e., combinations of T1- and proton density-/T2-weighted sequences) [38]. CL volume may be an additional outcome measure relative to WM lesion burden in clinical trials, given its potential for correlating with EDSS. Trials that employ neuroprotective measures or target cognition may specifically benefit from CL measurement. Furthermore, distinguishing specific CL types, such as leukocortical lesions, may improve correlations with cognitive impairment [37,39,40].

## 3. Iron-Derived Imaging and Chronic Inflammation

Iron is the most abundant trace metal in the brain, stored in oligodendrocytes and myelin, all features that make iron an attractive imaging contrast in MS [41,42]. In active MS lesions, dying oligodendrocytes release iron, which then accumulates inside astrocytes, microglia/macrophages, and axons, contributing to oxidative injury and mitochondrial dysfunction [43,44].

Four primary subtypes of MS lesions have been delineated: early active, chronic active (also known as mixed active/inactive, slowly expanding, or ‘smoldering’), inactive, and remyelinated (sometimes termed ‘shadow’) lesions [45,46]. Active lesions are characterized by hypercellularity and exhibit loss of myelin along with diffuse infiltration of blood-derived T cells and monocytes or microglia, predominantly localized in the perivascular zone and diffusing throughout the lesion area. Chronic active lesions are demyelinated and display a hypocellular lesion center surrounded by a rim of activated microglia/macrophages at the lesion border [46,47]. Inactive lesions are hypocellular, with only a sparse presence of T cells and microglia/macrophages within the lesion and lacking a macrophage/microglia rim. While all MS lesions can exhibit variable degrees of remyelination, approximately 20% of lesions demonstrate extensive remyelination, often referred to as shadow plaques [48,49].

A hallmark pathological feature of chronic active lesions is the presence of a rim comprising activated microglia and macrophages encircling an inactive lesion center with limited microglial and macrophage presence [50,51]. Within the phagocytes at the edge of chronic active lesions there is an accumulation of iron, which can be detected using susceptibility-based MRI techniques in vivo [52]. These techniques encompass susceptibility weighted imaging (SWI) phase, quantitative susceptibility mapping (QSM), or gradient echo (GRE)-derived R2/T2* maps, wherein the chronic active lesion rim manifests as a paramagnetic signal (hyperintense on QSM and hypointense on SWI (Figure 3) and T2* scans) with a linear or dot-like distribution at the lesion edge due to iron accumulation, hence termed paramagnetic rim lesions (PRLs) [53,54,55,56].

Remarkably, chronically active lesions with rims are more likely to continue expanding, whereas chronically inactive lesions do not, with rims being more common in both active relapsing MS and secondary progressive MS (SPMS) [57,58]. Iron accumulation is not confined to WM lesions, as it is also found in deep GM, correlating with disability, WM tract injury, and cognitive impairment [59].

### 3.1. Central Vein Sign

The central vein sign (CVS), visualized on susceptibility-sensitive MRI sequences, serves as a radiological biomarker in MS lesions, effectively differentiating MS from other conditions [60]. From a histopathological point of view, WM lesions in MS are characterized by inflammatory infiltrates that develop around venules [61]. These infiltrates, commonly referred to as perivascular cuffs, primarily consist of mononuclear cells. They exhibit dynamic accumulation, distribution, and evolution within the CNS, following recurrent waves of invasion from the peripheral blood [51].

CVS manifests as a central linear hypointensity within lesions, discernible on susceptibility-sensitive MRI sequences such as T2*-weighted scans. This hypointensity corresponds to the presence of the small vein or venule around which the lesion formed [62]. The CVS has garnered attention as a potential radiological biomarker, observed with a high frequency in MS lesions. A cut-off of 40% CVS positive lesions of the total amount of white matter lesions has been proposed as a biomarker able to discriminate between MS and non-MS conditions [63]. In a recent cross-sectional study [64] involving 1051 participants with variable inflammatory and non-inflammatory disorders, the adherence to the 40% CVS rule showed superior performance compared to the contribution of infratentorial, juxtacortical, and periventricular lesions in aiding the diagnosis of MS [64]. Other studies, instead, such as Maggi et al., used the 50% CVS cut-off to distinguish MS from vasculopathies [65]. Consequently, CVS is highly promising in effectively distinguishing MS from other conditions associated with nonspecific white matter lesions [64,65].

### 3.2. Paramagnetic Rim Lesions

Acute inflammatory lesions in the brain and spinal cord WM, characterized by a blood–brain barrier breakdown, are a hallmark of MS [66]. These lesions are observable on T1-weighted images following the administration of a gadolinium-based contrast agent. Acute inflammatory activity is closely linked with relapses and serves as a predictor of short-term disease outcomes [67].

Following the resolution of acute WM lesions, they may progress into chronic active lesions, become chronic inactive lesions, or undergo remyelination [68]. Chronic active lesions can be identified not only in the brain WM, but also in cortical regions and the spinal cord and represent focal sites of compartmentalized inflammation confined within the central nervous system, typically behind a relatively intact blood–brain barrier [67,69].

The identification of chronic inflammation in vivo, facilitated by noninvasive imaging techniques, is poised to underpin future endeavors aimed at tailoring treatments and stratifying patients for more targeted and effective clinical trials [52], focusing on progressive disease biology, using PRLs and slowly expanding lesions (SELs) as biomarkers of chronic inflammation [57,70]. 

PRLs, observed in a considerable proportion of individuals with MS, expand over time, signifying persistent inflammation and tissue destruction, making them relevant biomarkers for disease severity and progression [71]. PRLs are indicative of chronic inflammatory activity and are observed in a considerable proportion of individuals with MS, peaking in late relapsing MS and early SPMS [72]. 

PRLs exhibit a gradual expansion over time, surpassing the growth rate of non-PRLs [73]. They are estimated to manifest in approximately 40% of MS patients and, on average, constitute around 10% of the overall lesion count [58].

Studies such as those conducted by Absinta et al. have underscored the prognostic significance of chronic active lesions, particularly PRLs, in predicting clinical disease progression and their association with brain atrophy [52,74].

QSM, with its advantages over conventional imaging, holds promise for monitoring inflammation longitudinally and may serve as an intermediate outcome measure in clinical trials targeting iron metabolism or microglia in MS therapy [21]. Detection of rimmed lesions supplements established measures of activity without the need for exogenous contrast agents, potentially aiding in discriminating MS from other conditions. Iron imaging could represent a suitable clinical trial outcome measure, especially for therapies targeting iron metabolism or microglia [21,75].

The use of SWI, T2*, and QSM sequences is limited in the routine diagnostic and clinical trial evaluation protocols due to their prolonged acquisition times (i.e., 5–8, 20 min) and the scarce availability of 3T and 7T MRIs that greatly increase the possibility to detect CVS and PRLs [76].

## 4. Slowly Expanding Lesions

Slowly expanding lesions have been detected for the first time through an automated method, able to determine the volumetric deformation of lesions in a reference MRI scan in relation to follow-up scans through the use of the Jacobian determinant [77]. In the ORATORIO and OPERA trials, 72% of primary progressive MS patients and 68% of relapsing–remitting MS patients manifested at least 1 SEL [77]. Like PRLs, SELs have been quantified across all MS phenotypes, with a higher prevalence noted in progressive forms (especially secondary progressive) compared to relapsing–remitting MS. SELs contribute significantly to the total lesion burden, exhibiting a greater T1 hypointensity compared to non-SELs and displaying a higher likelihood of evolving into persistent black holes, suggesting an ongoing neuro-axonal damage. SELs also present a lower magnetization transfer ratio (MTR) and greater diffusivity in diffusion-weighted imaging (DWI), consistent with myelin loss and tissue destruction [78]. They have been associated with neurological motor and cognitive disability in MS [20,79]. Other investigations have revealed a notable correlation between EDSS progression and the proportion of SELs [52,79], suggesting SEL microstructural alterations as independent risk factors for disability accumulation and the conversion to secondary progressive MS (SPMS). SELs have been also described in relation to treatment. Fingolimod-treated patients showed lower volume and count of definite SELs compared to placebo [80], while BTK inhibitor Evobrutinib has been shown to reduce the rate of expansion of SELs compared to Dimethyl fumarate [81].

While both PRLs and SELs represent chronic active lesions, the nature of their relationship and their distinct associations with clinical outcomes remain unclear. In this regard, studies such as the one conducted by Calvi et al. have proposed that SELs outnumber PRLs in MS patients and that their concurrent occurrence is linked to more pronounced clinical progression [82].

## 5. Leptomeningeal Enhancement

Leptomeningeal enhancement (LME) represents a persistent enhancement of the leptomeninges, potentially indicating populations of immunologic cells contributing to ongoing neurodegeneration, cortical demyelination, and cortical atrophy observed in MS [83]. Notably, LME has been observed more with a higher prevalence in progressive forms of MS and it is correlated with global and cortical atrophy [23]. Post-contrast T2 FLAIR imaging has shown greater sensitivity in detecting enhancement compared to T1, primarily due to improved cerebrospinal fluid (CSF) contrast [84]. The association between LME and cortical subpial demyelination, perivascular macrophages, and T and B lymphocytes has been demonstrated in limited postmortem studies [83].

While LME is observed in approximately 25% of MS patients, it lacks specificity, as similar patterns are noted in other inflammatory/infectious and non-inflammatory/non-infectious conditions. The potential contribution of meningeal follicles to ongoing progression, particularly in progressive MS, warrants further investigation and validation as a potential outcome measure in clinical trials [85,86].

## 6. Brain Atrophy

Focal tissue loss in WM plaques is a significant contributor to brain atrophy in MS and longitudinal studies have highlighted the importance of early focal lesion volumes in predicting atrophy, suggesting inflammation as a crucial contributor to tissue loss in early disease stages [87]. As the disease progresses, additional mechanisms emerge, including microglia activation, meningeal inflammation, iron deposition, oxidative stress, and diffuse axonal damage in normal-appearing white matter (NAWM), independent of WM injury [88].

Whole-brain atrophy in MS progresses at a rate approximately three times faster than in healthy counterparts, particularly evident in the early stages of the disease, potentially impeding brain growth in pediatric-onset MS [87]. Its correlation with disability progression underscores its relevance as an outcome measure in therapeutic trials. However, criticisms persist regarding the slow rate of change, intra-subject variability, and disparities among atrophy software algorithms [89,90].

Regional atrophy studies have shown promise in providing insights into different pathogenic processes in MS [91]. Early volume loss of deep GM structures correlates strongly with the disease course and disability progression. Brain atrophy in MS affects various CNS structures, with ventricular enlargement being prominent in RRMS and cortical atrophy being more significant in progressive forms of the disease [87].

Regional measures of atrophy, particularly deep GM atrophy, have demonstrated potential as candidate measures for clinical trials, showing a correlation with disability progression and risk of secondary progression independent of other measures such as WM lesion volume [92,93]. Thalamic atrophy, notable even in CIS, has shown associations with the risk of clinically definite MS and cognitive impairment [94], making it a valuable marker for discriminating MS from other conditions such as neuromyelitis optica spectrum disorders (NMOSD) [21].

## 7. Choroid Plexus Enlargement

The choroid plexus (CP) plays a pivotal role in immune regulation within the CNS, serving as a crucial interface between the peripheral immune system and the CNS. Acting as a gateway, the CP facilitates lymphocyte entry from the peripheral blood into the CNS and contributes to CSF surveillance and antigen presentation [95].

Notably, significant enlargement of the CP is observed from the early stages of MS and is linked to heightened disease activity. This enlargement correlates with a higher relapse rate, increased volume of T2-hyperintense white matter lesions in the brain, greater inflammatory activity, and more severe disability progression [96,97].

According to some authors, CP involvement may be an early event in human demyelinating disease, as suggested also by murine models of experimental autoimmune encephalomyelitis (EAE), where the occurrence of inflammation within the CP precedes the emergence of parenchymal inflammatory infiltrates and the onset of demyelinating lesions in white matter [98,99].

Moreover, postmortem examinations of CP in MS patients have unveiled several key findings on the involvement of this structure, such as the detection of elevated levels of antigen-presenting cells within the CP stroma, the infiltration of peripheral leukocytes, the disruption of tight junctions in CP epithelium, and the endothelial overexpression of adhesion molecules implicated in lymphocyte migration [95,99].

Despite the wealth of evidence from translational and neuropathological studies, there remains a lack of in vivo demonstration of CP involvement in the inflammatory processes characteristic of MS [95,99].

Recent studies have shown that patients with MS exhibit significant CP enlargement compared to healthy controls, further supporting the notion that CP enlargement may represent an early phenomenon in MS, reflecting chronic CNS inflammation. Additionally, MS patients with cognitive impairment demonstrate higher CP volumes compared to cognitively preserved individuals, suggesting a potential association between CP enlargement and cognitive dysfunction. Furthermore, CP volume appears to serve as a possible biomarker for fatigue, as it is higher in fatigued MS patients compared to non-fatigued individuals [97].

## 8. Diffusion Tensor Imaging

Diffusion tensor imaging (DTI) is a powerful technique used to assess microstructural changes in tissue by analyzing the Gaussian diffusivity of water (Figure 4). Each voxel in DTI represents multiple diffusion-weighted acquisitions, modeled as a diffusion tensor containing three perpendicular eigenvectors with corresponding eigenvalues. In WM, diffusion is anisotropic, meaning it is not equally restricted in all directions and follows the direction of the axon. Key DTI measures include fractional anisotropy (FA), mean diffusivity (MD), axial diffusivity (AD), and radial diffusivity (RD) [100].

Changes in these metrics can identify subtle alterations in pre-lesional normal-appearing white matter (NAWM) and normal-appearing gray matter (NAGM), as well as lesional tissue [101].

In acute MS lesions, a reproducible pattern of decreased FA, increased MD, and increased RD is typically observed [102]. These findings are consistent with the higher water content, depletion of myelin and axons, and the occurrence of gliosis [100]. Changes in NAWM, such as decreased FA and increased MD and RD, can be detected at baseline in CIS and progress over time [103]. As for the progressive phases of the disease, the average MD in lesions, NAWM, and NAGM is higher in secondary progressive MS (SPMS) compared to primary progressive MS (PPMS) [104].

DTI can also detect changes in normal-appearing cortical and deep gray matter. Increases in MD in the NAGM of untreated RRMS patients over time, independent of brain atrophy, have been noted. Thalamic MD increases in MS patients correlate with clinical disability measures and lesion burden. Furthermore, DTI changes are apparent even in pre-lesional NAWM, preceding gadolinium enhancement by weeks to months [21].

DTI has the potential to discriminate the evolution of lesions and mechanistic differences. For example, FA decreases in major WM tracts following gadolinium enhancement, suggesting ongoing pathology. DTI metrics also correlate with clinical measures and specific cognitive domain impairments and can predict disability worsening, providing insights into functional networks and cognitive dysfunction in MS [105,106].

DTI has been used to evaluate therapeutic effects in small studies and clinical trials, showing promise in assessing treatment response and disease progression [107,108,109,110]. 

However, DTI lacks specificity in distinguishing between various pathological changes. For example, FA cannot differentiate among inflammation, demyelination, or leukoaraiosis. Consequently, advanced diffusion modeling methods have emerged to offer finer microstructural specificity. These methods include q-space imaging (QSI), diffusional kurtosis imaging (DKI), neurite orientation dispersion and density imaging (NODDI), high angular diffusion resolution imaging (HARDI), white matter tract integrity (WMTI), and multiple diffusion encoding [111,112].

Despite some limitations, such as the complexity of MS pathology and oversimplification associated with equating AD to axonal integrity, DTI remains a valuable tool for assessing microstructural tissue changes and could be a useful measure in clinical trials evaluating neuroprotective strategies [113]. With further MRI technology advancements, these techniques will likely become increasingly valuable in the clinical evaluation of MS patients.

## 9. Functional MRI

Functional magnetic resonance imaging (fMRI) represents a category of imaging techniques devised to illustrate regional, dynamic alterations in brain metabolism. These metabolic fluctuations may arise from task-evoked changes in cognitive states or spontaneous activities occurring in the resting brain [114]. Since its inception, fMRI has garnered extensive attention in numerous studies within the realms of cognitive neuroscience, clinical psychiatry/psychology, and pre-surgical planning [115]. The popularity of fMRI stems from its widespread accessibility (feasible on a clinical 1.5T scanner), noninvasive nature, relatively modest expense, and high spatial resolution [116].

fMRI leverages the diamagnetic and paramagnetic properties of oxyhemoglobin and deoxyhemoglobin to provide a noninvasive measure of blood oxygenation levels. This technique allows for the detection of activated areas in the brain during task-related fMRI and of functionally connected brain areas during resting-state fMRI (Figure 5), offering an insight into the correlation between clinical metrics and functional changes, such as reorganization, plasticity, and functional reserve, across different MS phenotypes [117].

fMRI stands out as a unique MRI modality capable of detecting functional changes alongside anatomical and pathological alterations. Studies indicate that individuals with MS exhibit increased activation and functional connectivity in various brain regions, potentially reflecting adaptive mechanisms. However, this increased synchronization in several brain regions during the early stages of MS may decline as the disease progresses, suggesting reduced reorganization due to tissue injury [118,119]. In contrast, during the more manifest stages of RRMS, increased connectivity and activation are observed, whereas later stages of the disease show decreased connectivity, often associated with cognitive manifestations [21].

Task-related fMRI studies have shown increased recruitment in cortical and thalamic regions in CIS and other MS phenotypes, highlighting its potential for discriminating between different disease stages [118,120]. Moreover, cognitive impairment in MS has been correlated with functional connectivity and activation differences observed using fMRI, particularly in regions associated with processing speed, executive function, and working memory [121].

Fatigue, a prevalent symptom in MS, has also been studied using fMRI, revealing unique hyperactivation patterns in certain motor attention networks associated with cognitive fatigue. Early detection of functional connectivity changes with fMRI provides a window for intervention before significant cognitive impairment occurs, offering potential avenues for cognitive rehabilitation trials [121,122].

## 10. Magnetization Transfer Imaging

Magnetization transfer (MT) imaging, another MRI technique, measures myelination and tissue damage by estimating interactions between free water protons and macromolecular protons. While conventional MRI techniques measure the relaxation times of the free pool, MT measures the exchange of magnetization between the bound pool, which is more abundant in myelin and axonal membranes, and the free water pool. The MT ratio (MTR), a semi-quantitative measure of the MT effect, reflects the number of macromolecular structures in a given tissue and is greater in the WM than in the GM [23,123]. In MS, MTR has been instrumental in detecting signs of WM damage beyond demyelinating lesions [124]. MTR values are lower in NAWM and NAGM compared to healthy controls and they are lower in T1-black holes than in T1-isointense, T2-visible lesions [124,125,126]. Microstructural changes detected by MTR are more noticeable around the ventricles, providing support to the theory of a CSF or ependymal-mediated pathogenesis [127]. Moreover, MTR decreases precede the appearance of new lesions and have shown good predictive potential for many outcome measures in MS, including long-term disability accumulation [128,129].

Studies have demonstrated temporal changes in MTR with decreases in pre-lesional tissue and recovery to baseline in subsequent months [130]. The evolution of gadolinium-enhancing lesions into CELs can be predicted by the short-term evolution of lesional MTR [131].

Several disease-modifying therapies, such as interferon-beta 1a, dimethyl fumarate, and natalizumab, have demonstrated promising effects on MTR, suggesting its potential for assessing microstructural damage and examining neuroprotective approaches in MS [132,133]. Consequently, this sequence has seen growing utilization in recent phase 2 and 3 clinical trials [134,135,136].

A recent phase 3 trial sub-analysis revealed that Siponimod improved brain tissue integrity in newly formed MTR lesions compared to placebo [137]. This highlights, once again, the potential of MT imaging for monitoring remyelination phenomena. 

The specificity of MT imaging is limited, because the signal can be influenced by water content and activated microglia. Additionally, MTR values vary depending on acquisition parameters and scanner types. These limitations represent a challenge to a clinical translation of this technique. Recent advancements in quantitative MT (qMT) aim to mitigate these discrepancies by offering more tissue-specific indices [138,139].

## 11. Proton Magnetic Resonance Spectroscopy

Proton magnetic resonance spectroscopy (MRS) is a valuable tool for providing quantitative information about the brain’s biochemical composition by analyzing the properties of different nuclei, such as hydrogen (1H), phosphorus (31P), carbon (13C), and sodium (23Na). Among these, proton (1H) MRS has been extensively used to characterize MS pathology since the early 1990s, offering insights into neuronal, axonal, and glial differences in both lesional and normal-appearing tissue [122,140].

Using characteristic shifts in proton resonance signals, metabolites can be identified, with their signal intensity reflecting their relative concentration. N-acetyl aspartate (NAA) stands as a crucial metabolite, recognized as a hallmark of neural integrity and functionality [141]. A reduction in the brain NAA/creatine (Cr) ratio indicates axonal damage, a phenomenon observed even in the earliest stages of MS such as those with radiologically isolated syndrome (RIS). Creatine (Cr), another cerebral metabolite, mirrors cellular metabolism and energy systems, serving as an indicator of changes in cell density and/or gliosis. Additionally, choline (Cho) serves as a marker of cellular membrane turnover, reflecting phospholipid synthesis and degradation. This metabolite correlates with the release of phospholipids during active demyelination. An elevation in the Cho/Cr ratio is associated with increased membrane cell turnover in active MS plaques, while a decrease in NAA levels is observed in inactive plaques as well as in NAWM [141,142,143].

MRS has shown promise in discriminating MS patients from healthy controls or those with other neuroinflammatory diseases like NMOSD. Moreover, it has been used to monitor treatment response in clinical trials with various disease-modifying therapies, such as interferon-β, glatiramer acetate, natalizumab, and biotin, among others. Despite its potential utility, challenges remain, including the need for standardization across centers, limitations in quantification methods, and partial volume effects in small lesions [144,145,146].

Sodium (23Na) MRS has also been explored in relation to MS, as it may reflect underlying pathological mechanisms such as demyelination and axonal injury. Elevated sodium concentrations in tissue have been observed in various MS phenotypes and correlate with disability measures [147].

## 12. Positron Emission Tomography

Positron emission tomography (PET) offers high specificity in exploring the mechanisms underlying brain pathological changes in MS. PET imaging with specific radiotracers allows for the quantification of neuroinflammation, demyelination/remyelination, and early neuronal damage, offering valuable insights into the pathophysiology of MS [148,149]. Tracers targeting the translocator protein (TSPO) have been widely used to detect innate immune cell activation [150], while other tracers target amyloid deposition, myelin, or glucose metabolism. PET imaging has revealed diffuse activation of innate immune cells in both WM lesions and cortical regions and has provided insights into disease evolution and lesion patterns not detectable by traditional MRI techniques [151,152,153,154].

Advancements in PET tracers and imaging techniques hold promise for bridging the gap between neuropathology and clinical studies in MS research. Longitudinal studies combining PET with advanced MRI may offer a deeper understanding of the pathological events underlying MS progression and disability [23]. Moreover, PET imaging has the potential to become a valuable tool for personalized MS medicine, providing insights into individual disease profiles and treatment responses [153,155,156].

## 13. Spinal Cord

Spinal cord imaging using MRI plays a crucial role in diagnosing MS, assessing risk in individuals with RIS or CIS, and monitoring disease activity during treatment [157,158,159]. However, imaging the spinal cord presents challenges due to its small size, physiological motion, and surrounding bone structures [160]. Despite these difficulties, efforts have been made to establish standardized clinical acquisition parameters for spinal cord imaging, including sagittal and axial planes with various sequences such as T2-weighted, short T1 inversion recovery (STIR) or DIR, and post-contrast T1-weighted sequences. Incorporating axial views and specific sequences like T2-weighted with STIR or DIR can enhance lesion detection in the spinal cord [157,161]. Upper spinal cord regions (e.g., C1–C2, C1–C3, and C2–C3) are less affected by breathing-related artifacts [162,163] and are used for studying the spinal cord and its atrophy.

Spinal cord atrophy correlates with disability, especially in progressive forms of MS, making it an attractive outcome measure. Additionally, GM lesions in the spinal cord have been found to correlate with disability and may be more readily detectable than cortical lesions [164].

Spinal cord atrophy, calculated by a reduction in the cross-sectional area of the spinal cord over time, represents a prominent and clinically relevant aspect of MS and its subtypes [159]. In fact, in cases of clinically definite MS, the rate of cord atrophy typically ranges between 1% and 5% annually, with notably higher rates observed among patients with progressive forms of the disease [165]. The emergence of cord atrophy is considered to be a primary substrate contributing to the accumulation of disability, potentially contributing to up to 77% of disability progression over a five-year period [162,163].

Moreover, recent studies have highlighted the clinical relevance of spinal cord atrophy, demonstrating significant correlations between the annual rate of spinal cord atrophy and disability progression. Additionally, a smaller spinal area and elevated baseline spinal cord lesions have been identified as independent predictors of disability over the subsequent 5 years [166].

Advanced imaging techniques like DTI have been applied to the spinal cord, allowing for the assessment of microstructural changes in both normal-appearing and lesional regions. DTI measures have shown correlations with histological markers of demyelination and have been useful in tracking disease progression longitudinally. Other methods, including myelin water fraction (MWF), MRS, MT, and fMRI, have also provided insights into myelination, axonal density, and functional changes in the spinal cord of MS patients [167,168].

## 14. Magnetic Resonance Fingerprinting

Magnetic resonance fingerprinting (MRF) represents a pioneering technique in quantitative MRI, enabling the assessment of diverse tissue properties within a single, expedited acquisition. MRF operation hinges upon the intentional modulation of MR system parameters, eliciting distinct evolutions from each tissue type, commonly referred to as “fingerprints” [169,170].

By matching these tissue fingerprints with a comprehensive database, specific tissue properties can be accurately delineated. Clinically, MRF has demonstrated utility in discerning between different types of brain tumors and has exhibited promise in quantifying brain pathology in MS [21,171,172].

Continual advancement and validation of MRF technology, along with exploration of novel clinical applications, remain ongoing as dynamic endeavors in the field [21].

## 15. Conclusions

In conclusion, MRI has a pivotal role in the diagnosis and follow-up of multiple sclerosis in all the phases of the disease. Advanced MRI techniques for brain and spinal cord imaging in MS are rapidly evolving and hold great potential as noninvasive biomarkers for inflammation and neurodegeneration. Nevertheless, all the described MRI techniques, despite their great potential, have some limits, mainly related to a high acquisition time, elevated costs, or lack of reproducibility (as illustrated in Table 1), that make difficult their everyday use in clinics. Ideally, in the future, the combination of one or more of these techniques could be an important tool for clinicians in the diagnosis, evaluation, and prediction of the phase of the disease, and in evaluating treatment efficacy, giving a better and more complete view of the disease from its early stages.

## Figures and Tables

**Figure 1 diagnostics-14-01120-f001:**
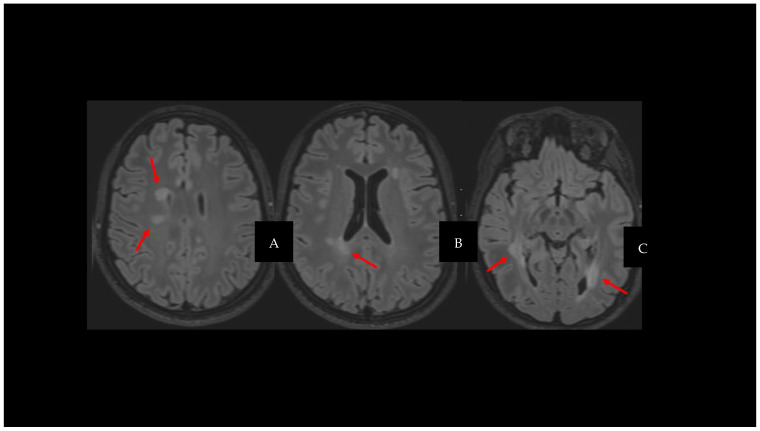
Fluid-attenuated inversion recovery (FLAIR), axial plane (**A**–**C**); red arrows show white matter lesions.

**Figure 2 diagnostics-14-01120-f002:**
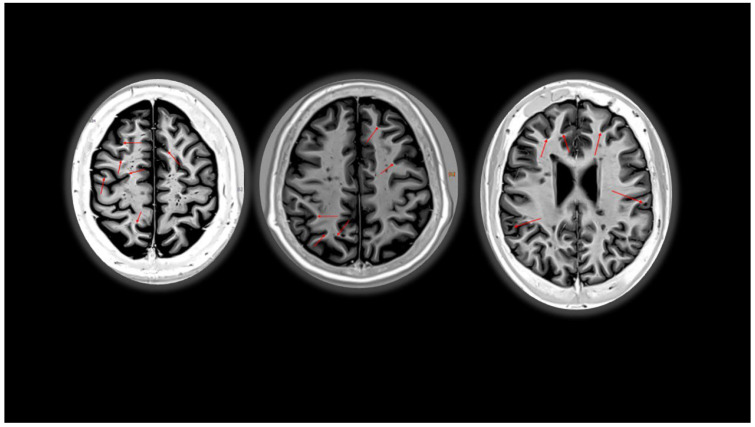
Phase-sensitive inversion recovery (PSIR), axial plane. Red arrows show cortical lesions.

**Figure 3 diagnostics-14-01120-f003:**
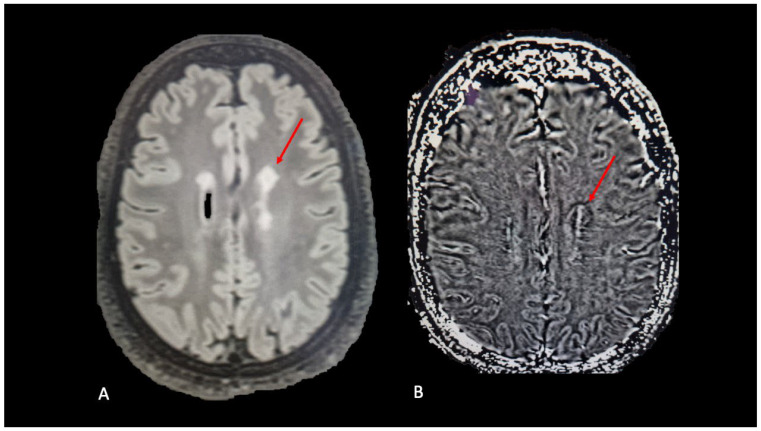
(**A**) Fluid-attenuated inversion recovery (FLAIR), axial plane; red arrow shows white matter lesion. (**B**) Susceptibility weighted imaging, phase image, axial plane; red arrow shows hypointense rim at the edge of the lesion.

**Figure 4 diagnostics-14-01120-f004:**
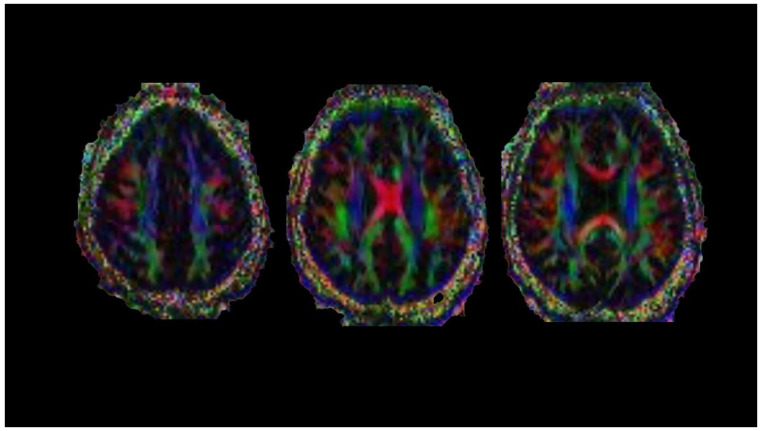
Diffusion tensor imaging (DTI), axial plane. The convention for color coding is as follows: red = transverse fibers; green = anteroposterior fibers; blue = craniocaudal fibers.

**Figure 5 diagnostics-14-01120-f005:**
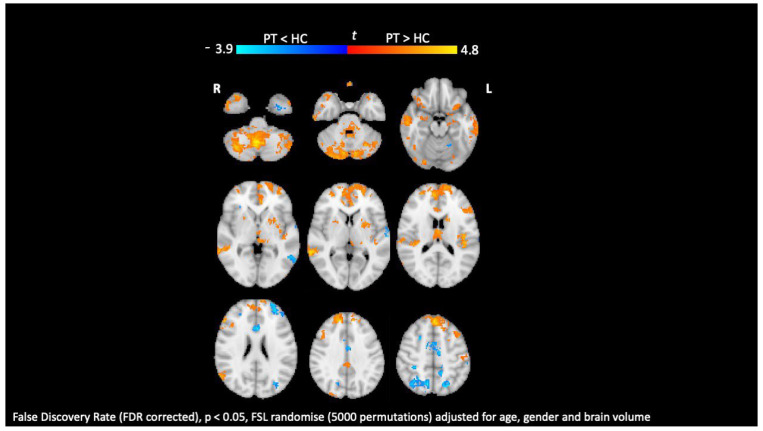
Resting-state insular functional connectivity. Comparison between patients with MS (PT) and healthy subjects (HC). Yellow–red indicates increased insular functional connectivity; light-blue–blue indicates reduced insular functional connectivity. False discovery rate (FDR corrected), *p* < 0.05, FSL randomize (5000 permutations) adjusted for age, gender, and brain volume.

**Table 1 diagnostics-14-01120-t001:** Main advantages and disadvantages of decsribed Imaging Techniques.

Imaging Techniques	Novel Finding	Clinical Advantages	Disadvantages
DIR, PSIR, MP2RAGE	CLs	Marker of disability and cognitive impairment	Acquisition time, availability of 3T and 7T MRI for better resolution
SWI, T2*, QSM	CVS, PRLs	Differential diagnosis (CVS); disability and disease progression (PRLs)	Availability of 3T and 7T MRI for better resolution, acquisition time
DTI	Pre-lesion alteration in NAWM	Early detection of new lesions	Lack of specificity
fMRI	Functional changes, plasticity, functional reserve	Study of fatigue and functional reserve	Lack of standardized protocol with high inter-subject variability
MT imaging	Pre-lesion alteration	Predicting long term disability accumulation	Lack of specificity and lack of standardized protocol
MRS	Neural integrity and functionality	Differential diagnosis	Lack of reproducibility and lack of standardized protocol
PET	Quantification of neuroinflammation	Quantification of demyelination and remyelination	Radiations

DIR: double inversion recovery; PSIR: phase-sensitive inversion recovery; MP2RAGE: rapid acquisition with gradient echo; SWI: susceptibility weighted imaging; T2*: gradient echo (GRE) derived R2/T2* maps; QSM: quantitative susceptibility mapping; DTI: diffusion tensor imaging; fMRI: functional MRI; MT imaging: magnetization transfer imaging; MRS: proton magnetic resonance spectroscopy; PET: positron emission tomography; CLs: cortical lesions; CVS: central vein sign; PRLs: paramagnetic rim lesions; NAWM: normal appearing white matter.

## Data Availability

Data are available in the article and in the references.

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
