# Peer review of "Advanced MRI Techniques: Diagnosis and Follow-Up of Multiple Sclerosis"

_diagnostics, 2024, doi:10.3390/diagnostics14111120_

Round 1

Reviewer 1 Report

Comments and Suggestions for Authors

The manuscript entitled Advanced MRI techniques: diagnosis and follow up of Multiple Sclerosis brings information about the importance of MRI in MS diagnosis.

Observations

Introduction

In Introduction please make a clinical classification of MS and enumerate all the diagnosis tests, including CSF examination. There are situation when CSF examination is needed for a precise diagnosis. 

Please also specify the role of inflammation in the pathogenesis of MS and the connection between inflammation and specific signs in MRI examination for MS diagnosis. Please make the difference between new active lesions and the old ones. 

- line 55 you wrote MS and in line 79 you wrote multiple sclerosis. Please be consistent with abbreviations in all the manuscript.

At the end of Introduction please specify clear the aim of this paper.

- line 166 - what do you mean by tissue destruction. Please specify.

- line 249 - please reformulate the title here " Choroid Plexus" in "Choroid plexus enlargement" (eventually in MRI examination).  Did you find an anatomopathological correspondence ?

- line 386 reformulate the subtitle because your manuscript is not refering at PET.

- line 415 - the same observation.

Please make 1 or 2 tables with the correlations between specific imaging characteristics and clinical forms of MS and eventually with specific symptoms such as cognitive decline. This is because the authors describe these details in text, and would be useful to be organized in tables. 

Please write separate chapters about FLAIR and SWI examinations after or before DTI.

Reviewer 2 Report

Comments and Suggestions for Authors

The review chose a title that emphasizes that it focuses on imaging techniques in MS; but the content is not compatible with the title. It is seen that the imaging findings of MS are also included in the article content. In this respect, the article has a complex structure and should be completely reorganized. First of all, the imaging findings and criteria used in the diagnosis of MS should be explained, and since a review focusing on imaging is presented, each finding should be presented with imaging examples. Afterwards, advanced imaging methods should be presented under separate headings, together with imaging examples, highlighting their contribution to existing diagnostic criteria. McDonald criteria still play an important role in the diagnosis of MS. This part should be included in more detail.

As far as I can see from the author details, there are no radiologists among the authors. An article discussing imaging features should include an expert in the field of imaging. Imaging findings are described as amateurish, radiologist support is required.

Reviewer 3 Report

Comments and Suggestions for Authors

As a reviewer, I appreciate the comprehensive nature of this review article on advanced MRI techniques used in the diagnosis and follow-up of multiple sclerosis (MS). The authors have provided a thorough overview of various imaging modalities and their potential as non-invasive biomarkers for inflammation and neurodegeneration in MS.

The article is well-structured, with clear descriptions of each advanced MRI technique, including cortical lesion imaging, paramagnetic rim lesions, slowly expanding lesions, central vein sign, leptomeningeal enhancement, brain atrophy, diffusion tensor imaging, functional MRI, magnetization transfer imaging, proton magnetic resonance spectroscopy, positron emission tomography, spinal cord imaging, and magnetic resonance fingerprinting. The authors have effectively explained the principles behind each technique and their clinical relevance in the context of MS, supporting their claims with appropriate references to recent studies and clinical trials.

I want to highlight some additional limitations of this article that the authors should consider addressing:

  1. Lack of essential analysis: While the article gives a complete evaluation of various superior MRI techniques, it needs a more critical evaluation of the research referred to. The authors must speak about the strengths and weaknesses of the essential studies of things assisting every technique and highlight any conflicting findings or inconsistencies within the literature.
  2. Limited discussion on clinical implementation: The article would benefit from an extra precise dialogue on the practical factors of imposing those advanced MRI techniques in clinical settings. This need to encompass records of the availability of these techniques, the want for specialized gadgets or expertise, and the value-effectiveness of incorporating these modalities into habitual medical exercise.
  3. Limited Discussion on Cost-Effectiveness: The review must address the cost-effectiveness of implementing advanced MRI techniques in routine clinical practice. Including a discussion on the economic impact, such as cost-benefit analyses, would be beneficial, especially for healthcare settings with limited resources.
  4. Omission of Technical Challenges: The technical challenges associated with each MRI technique, such as the need for high-field MRI systems (e.g., 3T or 7T), which are not universally available, should be thoroughly discussed. The practical barriers to adopting these advanced techniques, such as installation costs, maintenance, and required technical expertise, should be clearly outlined.
  5. Generalization of Results: The article occasionally generalizes results from advanced MRI techniques without sufficient differentiation between different MS subtypes or stages of the disease. A more nuanced discussion of how these imaging techniques perform across various MS phenotypes would provide a clearer picture of their utility.
  6. Insufficient emphasis on standardization: The authors briefly point out the need for standardization across facilities for strategies like proton magnetic resonance spectroscopy. However, the article needs to offer a closer dialogue on the significance of standardization for all advanced MRI strategies, which include the development of steady acquisition protocols, publish-processing strategy, and reporting suggestions.
  7. Lack of comparative analysis: The article offers every superior MRI method for my part without supplying a comparative study in their relative strengths, boundaries, and scientific software. A section evaluating the various strategies and their suitability for exceptional aspects of MS prognosis and monitoring would be treasured by readers.
  8. Limited patient angle: The article mainly focuses on the technical components of superior MRI techniques and their ability as biomarkers for MS. However, it would be helpful to include a short dialogue with the affected person to reveal the acceptability of these strategies, potential discomfort or aspect effects, and the importance of patient training and conversation while enforcing these modalities in clinical practice.
  9. Need for more discussion on multi-modal strategies: The topic discusses each superior MRI technique separately; it would be worth including a section on the potential advantages of combining multiple modalities for a complete evaluation of MS pathology. This should include a dialogue on the complementary nature of various strategies and the capacity to grow multi-modal imaging biomarkers.
  10. Limited discussion on translational research: The article may want to take advantage of a more designated discussion of the translational aspects of those advanced MRI techniques, which include their capacity for informing the improvement of novel healing techniques and facilitating drug discovery efforts in MS.
  11. Underrepresentation of Alternative and Complementary Techniques: The article could improve by discussing how advanced MRI techniques compare or integrate with other diagnostic methods. For instance, how do these techniques perform compared to CSF analysis, optical coherence tomography (OCT), or other neurophysiological tests?

Round 2

Reviewer 2 Report

Comments and Suggestions for Authors

Revised version is more acceptable

Reviewer 3 Report

Comments and Suggestions for Authors

The manuscript has been significantly improved.